# Digital Planning for Immediate Implants in Anterior Esthetic Area: Immediate Result and Follow-Up after 3 Years of Clinical Outcome—Case Report

**DOI:** 10.3390/dj11010015

**Published:** 2023-01-03

**Authors:** Saulo Henrique Salviano, João Carlos Amorim Lopes, Igor da Silva Brum, Kelly Machado, Marco Tulio Pedrazzi, Jorge José de Carvalho

**Affiliations:** 1Implant Dentistry Department, São Leopoldo Mandic University, Brasilia 13045-755, Brazil; 2Implant Dentistry Department, Dental School, Portuguese Catholic University, 1649-023 Viseu, Portugal; 3Implant Dentistry Department, State Faculty of Rio de Janeiro, Rio de Janeiro 20550-013, Brazil; 4Department of Histology and Embryology, State University of Rio de Janeiro, Rio de Janeiro 20551-030, Brazil

**Keywords:** guided surgery, implants, connective tissue, biomaterial, bone, esthetic

## Abstract

In this case report, we demonstrate how the correct positioning of implants, associated with optimal gingival conditioning, and the correct choice of biomaterial can yield very predictable and fantastic aesthetic results. Objective: We aimed to use dental implants to rehabilitate the area of elements #11 and #21 in a satisfactory surgical and prosthetic manner, using guided surgery, connective tissue, nano-biomaterials, and a porcelain prosthesis. Case Report: A 32-year-old male patient presented with bone loss of elements #11 and #21, which was proven radiographically and clinically. Thus, oral rehabilitation with the use of dental implants was required. It was decided to proceed via digital planning with the DSD program (Digital smile design) and with the software Exoplan, (Smart Dent-Germany) whenever it was possible to plan immediate provisional and accurate dental implant positioning through reverse diagnostics (Software Exoplan, Smart Dent-German). The dental elements were extracted atraumatically; then, a guide was established, the implants were positioned, the prosthetic components were placed, the conjunctive tissue was removed from the palate and redirected to the vestibular wall of the implants, the nano-graft (Blue Bone^®^) was conditioned in the gaps between the vestibular wall and the implants, and, finally, the cemented provision was installed. Results: After a 5-month accompaniment, an excellent remodeling of the tissues had been achieved by the implants; consequently, the final prosthetic stage could begin, which also achieved a remarkable aesthetic result. Conclusions: This report demonstrates that the correct planning of dental implants, which is associated with appropriate soft tissue and bone manipulation, allows for the achievement of admirable clinical results.

## 1. Introduction

Dental implants have undergone enormous changes in several areas of their development, among which we can cite the two most important: (1) the improvement of the design of implants, which has enabled higher locking even in bone types 3 and 4 [1], and (2) the advances in the surface technology of dental implants, which have improved the interaction between titanium and the cells responsible for the osteointegration process, along with increasing the surface contact area. These concepts have enabled a change in surgical perspectives, as it is now possible to plan the installation of implants together with temporary or definitive prostheses, thus reducing treatment time and leading to more predictable results [2].

Among other such advances, we can mention that the discovery of computed axial tomography (today known as computed tomography) and the development of new interactive software allow for almost realistic virtual planning, with the aim of guiding a given surgery precisely towards a specific target. These targets can be different organs, dental implants, grafts, and even soft tissues, which greatly improves surgery in general oral surgery specifically [3,4]. The possibility of the immediate installation of dental implants provides a more reliable alternative regarding the response of soft tissues and bones surrounding the regenerated area. In this way, reverse planning with the aid of software has facilitated the creation of provisional materials that are more adapted to the receiving beds. Additionally, this strategy creates a better fit, reduces working time, and, consequently, reduces a patient’s physical tissue stress when compared to the implementation of provisional materials without digital planning [5].

The use of connective tissue has become a common clinical practice in aesthetic reconstructive surgery. This technique, which consists of removing connective tissue from the palate, has proven to be an excellent complement to the procedure performed. This is because it allows the surgeon to achieve greater short-term and long-term predictability of tissue stabilization in rehabilitation [6].

For years, much has been discussed concerning the need to use biomaterials in the gaps formed between the implant areas and the remaining vestibular wall when immediate implants are chosen. Today, it can be safely stated that, regardless of the size of this gap, it is extremely necessary to fill it with some type of biomaterial, whether with autogenous bone, xenogenous grafts, or, more recently, with alloplastic (synthetic) nanoparticle grafts [7,8,9].

The short and long-term health of the implants is directly related to the patient’s oral health, so it is very important to perform prophylaxis before surgery because the peri-implant microbiota possesses lower microbial quality than the periodontal microbiota; however, it can be immensely more complex and progress from peri-implant mucositis to peri-implantitis more quickly [10].

The objective of this case report was to demonstrate a method for obtaining predictable and extremely favorable results in the short and long term using traditional bone and tissue (connective) reconstructive techniques that have been enabled by the high-performance digital computing technologies currently available.

## 2. Materials and Methods

### 2.1. Patient Data

A 32-year-old male who was a non-smoker, clinically fit (no systemic disease), and did not present bruxism presented to the Department of Oral and Maxillofacial Surgery at Faculdade de Medicina e Odontologia São Leopoldo Mandic (SLM, DF, Brazil) in January 2019. The patient’s main complaint involved his fractured maxillary incisors with total loss of two crown structures—#11 and #21—that had suffered recent prosthetic losses, thus rendering the patient functionally and aesthetically impaired. Medical data and complementary exams confirmed normal health conditions. Cone Beam Computed Tomography (CBCT) confirmed the clinical findings of elements #11 and #21 having extensive caries, in addition to significant bone loss. The patient gave his informed consent to publish the details of the case and any accompanying images (Figure 1, Figure 2 and Figure 3).

### 2.2. Treatment Planning/Execution

The treatment began with the prior digital planning for the construction of the surgical guides, followed by the extraction of elements #11 and #21, installation of the implants, application of the biomaterial, positioning of the connective tissue, and finally the creation of provisional crowns on the previously installed implants. After about six months of the initial treatment, the second phase of the treatment was planned, which consisted of the installation of the ceramic crowns. In view of the patient’s consent (signed authorization) to participate in this treatment, which had been previously provided, the clinical case was initiated in accordance with the bioethical guidelines that govern SLM (DF, Brasilia, Brazil).

### 2.3. Biomaterials and Dental Implants

For bone reconstruction surgery, the alloplastic nano-biomaterial Blue Bone^®^ (Regener^®^, Curitiba, Brazil) was used. The implants used were a 3.5 × 13 mm morse taper of the Avantt model (Systhex^®^, Curitiba, Brazil).

### 2.4. Virtual Planning

The ideal positioning of the implants was planned using the Intraoral Scanner (Carestan 3600, Atlanta, GA, USA) of Exocad and the software Exoplan, Smart Dent-Germany and DSD (Digital Smile Design, WellClinic), and through DICOM (Digital Imaging and Communications), it was possible to perform the virtual reconstruction of the regions of elements #11 and #21 three-dimensionally and to determine the best positioning of the implants according to the anatomical landmarks present in these regions. At the end of the virtual surgical planning procedure, a virtual surgical guide was created and, consequently, the definitive clinical surgical guide that was used at the time of the surgery for the initial and final drilling and the installation of the implants, shown in Figure 4 and Figure 5.

### 2.5. Surgical Procedures

All surgical procedures, including follow-up assessments, were performed at the Department of Oral and Maxillofacial Surgery at the clinical facilities of SLM (DF, Brazil).

Amoxicillin (Amoxil^®^, GlaxoSmithKlein, Brentford, UK) (2 g, O.I.) was administered 1 h before surgery, together with Chlorhexidine Cluconate (0.12%) (Periogard^®^, Colgate, New York, NY, USA). Mouthwash was provided immediately before surgery.

Local anesthesia was induced by injection of Lidocaine Hydrochloride (2%) and Epinephrine 1:100,000 (Alphacaine 100^®^, DFL, Rio de Janeiro, Brazil). The dental elements were extracted in the least traumatic way possible, respecting the soft and bone tissues. Then, the surgical guide was fixed, and the implants installed, i.e., Cone Morse (Systhex^®^, Curitiba, Brazil). Consequently, the guide was removed, and the prosthetic trunnions were installed, both cemented to 3.3 × 6.0 × 2.5 m. From this stage, tissue manipulation began, wherein a fragment of connective tissue from the palate was removed and positioned in the buccal region of elements #11 and #21. The gaps between the implants and the remainder of the buccal plate were immediately filled with the nano-biomaterial composite Nano-HA/β-TCP (Blue Bone^®^, Regener^®^, Brazil) and blocks of provisional material were milled and installed (Figure 6, Figure 7, Figure 8, Figure 9, Figure 10 and Figure 11).

## 3. Results

### Result and Follow-Up

After the 6-month osseointegration period, the total adaptation of the tissues involved in the regeneration was observed, presenting great stability and satisfactory aesthetics. Finally, after clinically testing the stabilities of the implants, the final prostheses were made with hand-stratified ceramics (Noritake EX-3 porcelain, Tokyo, Japan), the occlusion was tested with carbon using laterality movements and contact points, and the adaptation of the cervical margin and the optimal aesthetic conditions were achieved. The results were highly satisfactory and approved by the patient. The patient was followed-up monthly for the next 12 months, wherein we assessed the implants’ stability, possible bone loss, gingival health, and occlusal changes (Figure 12, Figure 13, Figure 14 and Figure 15).

## 4. Discussion

Several authors have claimed that the interaction between the most advanced and sophisticated digital aspects has drastically changed the medical sector, especially dentistry. This combination of technological applications can improve oral health care, facilitate workflow, shorten delivery time, minimize error margins, and, above all, enable unprecedented aesthetic levels [11,12,13]. We agree with this notion, because with the help of the advances in technology it was possible to plan, execute, and finalize the presented case in an aesthetically acceptable way.

Reverse planning is, by many, ignored, but in the literature, there are numerous studies where the correct use of this resource drastically reduces the failure rate in the initial phase of rehabilitation [14,15,16]. The reverse treatment was an excellent resource for the satisfactory clinical results of this study.

In the case report presented, a fragment of connective tissue removed from the palate was used to improve the gingival conditioning of the regenerated area. The use of these surgical techniques is widespread and the techniques themselves are present in several variations. Through 3D diagnosis, it is possible to determine the optimal location to remove the fragment with the help of a guide, thus reducing the chance of committing a significant anatomical accident [17].

Volumetric changes after connective tissue grafting around newly installed dental implants have already been evaluated [18,19,20]. For cases in which no type of tissue control was used (palate connective graft or porcine membrane), there was, after 6 months, 100% gingival retraction. The areas in which tissue was manipulated and that used the porcine membrane obtained a volume loss of 80%. Finally, for cases with manipulations derived from the palate, the percentage of loss was 50% [21].

Levine et al., considered that there are 10 keys (steps) by which to succeed in the surgery of immediate implants positioned in the aesthetic zone [22], namely, two planning keys for the treatment, five surgical keys, and three prosthetic keys. Together, they aim to minimize soft and hard tissue complications for the optimal aesthetic restoration of the implant. We can consider that the clinical case presented herein followed the ten keys because we employed two stages of planning (tomographic and reverse), five surgical stages (atraumatic exodontia, tissue manipulation, the installation of biomaterial, guided surgery, and the correct positioning of the implants), and three prosthetic stages (the confection of the milled provisional material, the design of the porcelain prostheses, and digital prosthetics planning).

Guided surgery is an excellent alternative when the amount of bone in the patient is large enough to receive dental implants without the need for an extensive surgical cut [23]. This technique provides surgical predictability that provides a great deal of security to the surgeon and comfort to the patient [24]. In the case report presented herein, we can see that guided surgery was an excellent surgical alternative, wherein the patient’s clinical response was extremely satisfactory, enabling the manufacture of an aesthetically optimal porcelain prosthesis.

Reverse planning is a very important resource for the prosthetic result to be as faithful as possible to the adjacent teeth or even to the total prosthesis on implants [25]. When we examined at the case presented herein, it was possible to see that reverse planning worked very well, that the provisional prostheses were aesthetically well-adapted, and that it was possible to create an ideal functional space for the fabrication and adaptation of the definitive porcelain prostheses.

The maintenance of periodontal tissues is related to the use of correct surgical techniques, a well-adapted definitive prosthesis, and mainly to the bacterial control. A group of researchers investigated the effectiveness of chlorhexidine in relation to postbiotics in 20 patients, showing that both agents significantly decreased the levels of peri-implant mucositis [26] These findings are quite relevant as they indicate that without adequate oral health control as presented in this clinical case, the chances of developing peri-implant mucositis are very high, and may lead to aesthetic and functional failure.

These results are extremely interesting and were confirmed in the case presented, wherein a larger connective tissue fragment was introduced in accordance with the resorption rate. The aesthetic result of the tissue-remodeling procedure with respect to the regenerated region was very favorable and is in line with the extensive literature regarding the success of an aesthetically favorable outcome attained through the use of guided surgery, software for reverse planning, surgical dexterity, and porcelain prosthetics [27,28,29].

## 5. Conclusions

Cases of aesthetic implant placement require special consideration as well as the use of a combination of techniques to ensure the achievement of pink and white aesthetics. In the clinical case presented herein, we employed the combination of the following factors: (a) the correct planning of the dental implants; (b) the precise manipulation of soft tissues; (c) the choice of an effective biomaterial; and (d) the use of new technologies associated with planning and surgical and prosthetic execution. By following these guidelines, it was possible to grant the patient a predictable and long-lasting aesthetic and functional outcome.

## Figures and Tables

**Figure 1 dentistry-11-00015-f001:**
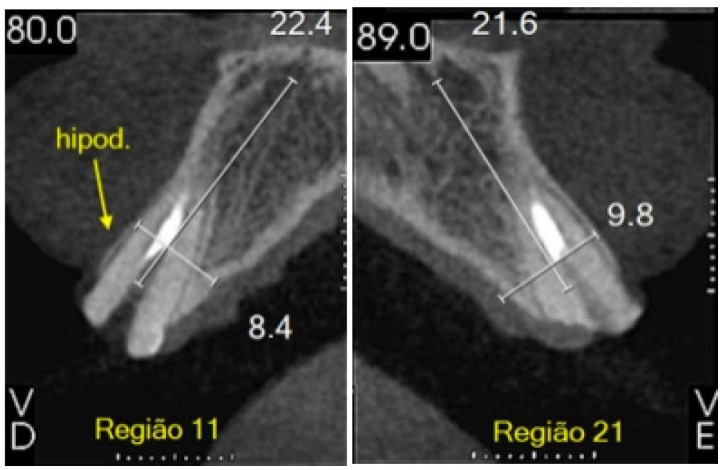
Initial tomography.

**Figure 2 dentistry-11-00015-f002:**
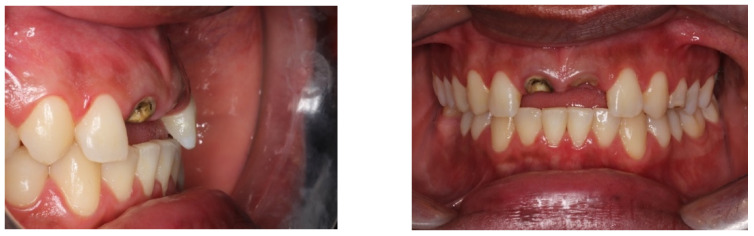
Front and side photo of elements #21 and #11.

**Figure 3 dentistry-11-00015-f003:**
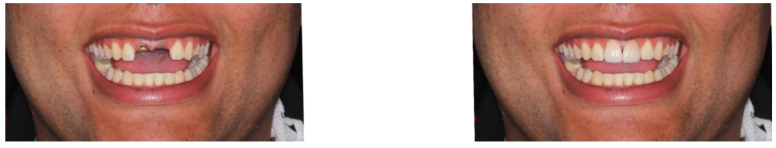
Reverse treatment photo of elements #21 and #11 via DSD program.

**Figure 4 dentistry-11-00015-f004:**
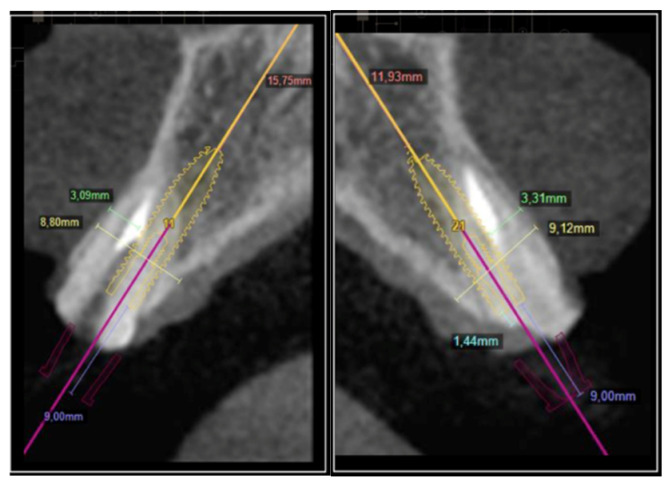
Planning the reconstruction of elements #11 and #21 in the DSD program.

**Figure 5 dentistry-11-00015-f005:**
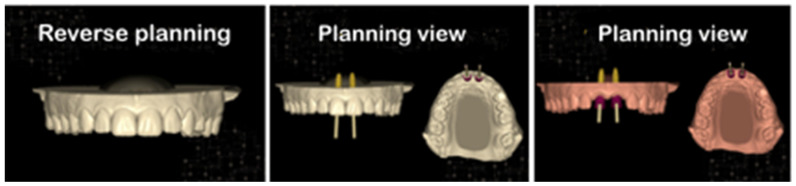
Reverse planning, evaluating implant positioning, and provisional design via DSD program.

**Figure 6 dentistry-11-00015-f006:**
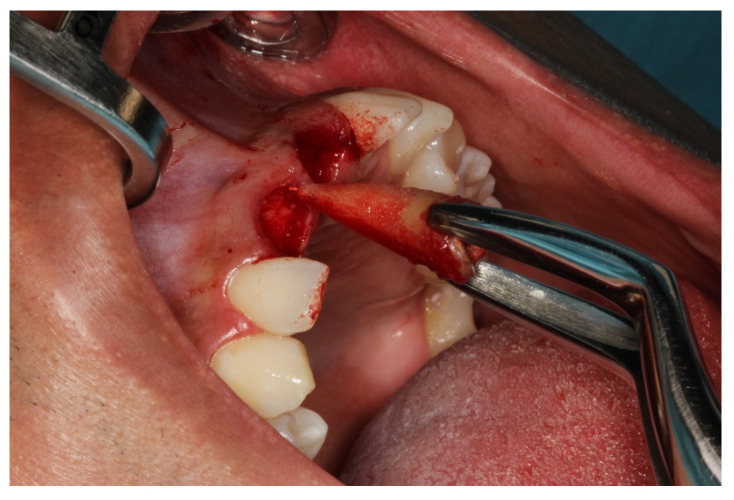
Atraumatic extraction of elements #11 and #21.

**Figure 7 dentistry-11-00015-f007:**
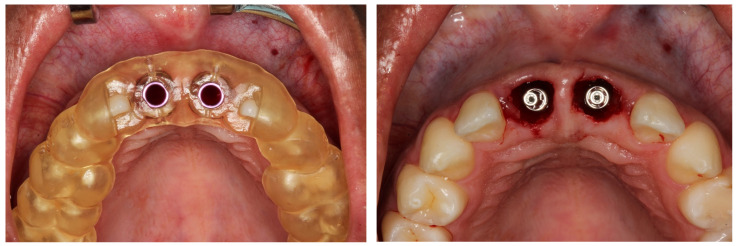
Installation of the surgical guide and Implants installed in the correct positioning and installation of cemented components.

**Figure 8 dentistry-11-00015-f008:**
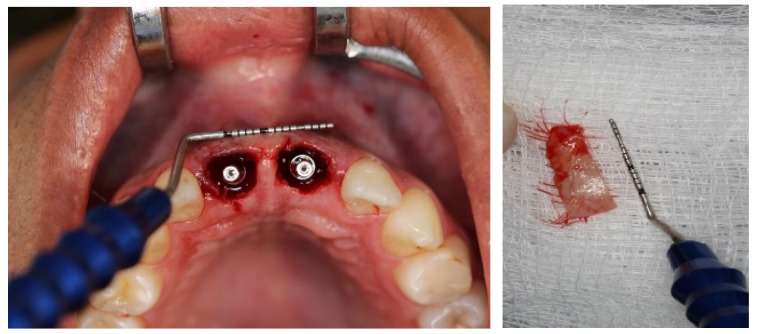
Measurement and removal of the connective tissue fragment.

**Figure 9 dentistry-11-00015-f009:**
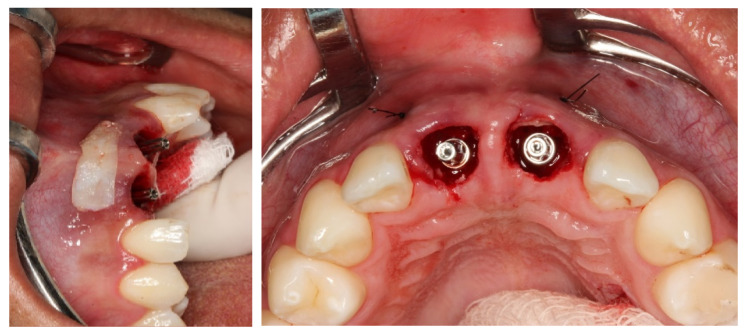
Connective tissue’s placement and installation.

**Figure 10 dentistry-11-00015-f010:**
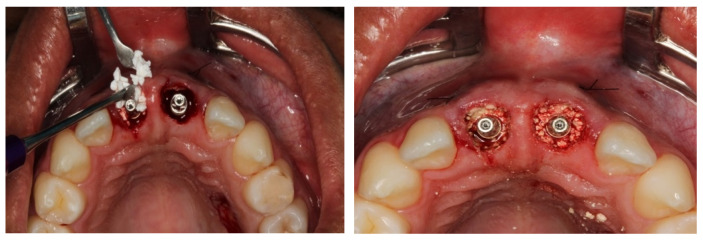
Filling of the gaps with the nano-biomaterial (Blue Bone^®^).

**Figure 11 dentistry-11-00015-f011:**
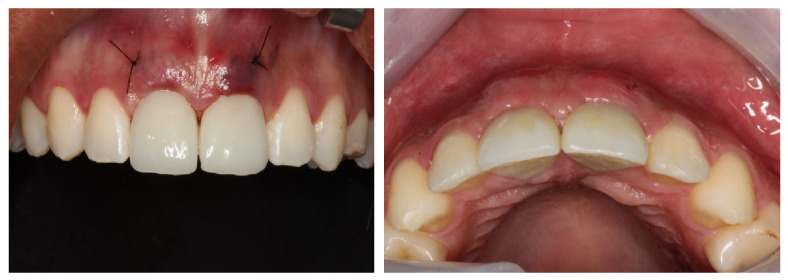
Surgical follow-up after 7 days, showing total tissue preservation.

**Figure 12 dentistry-11-00015-f012:**
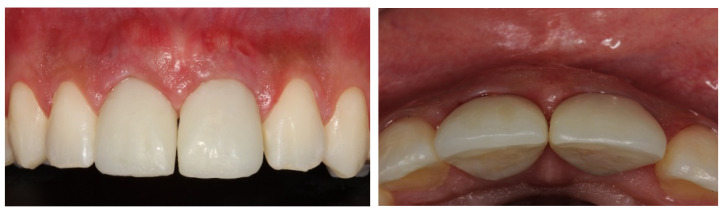
Surgical follow-up after 5 months, showing tissue remodeling and perfect adaptation of the provisional material.

**Figure 13 dentistry-11-00015-f013:**
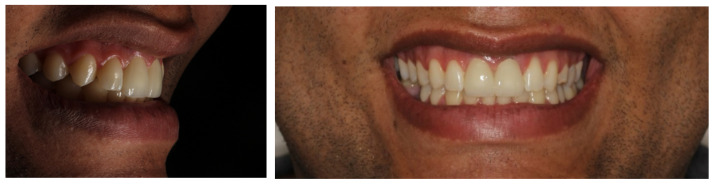
Prosthetic finish completed with veneered porcelain-cemented crowns.

**Figure 14 dentistry-11-00015-f014:**
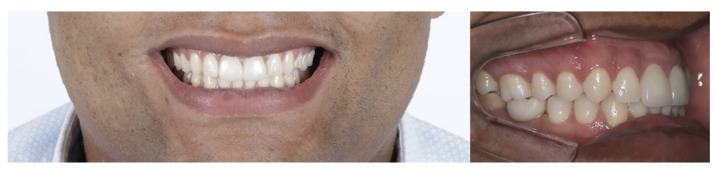
Frontal and lateral view of 3-year follow-up, showing the permanence of tissue stability.

**Figure 15 dentistry-11-00015-f015:**
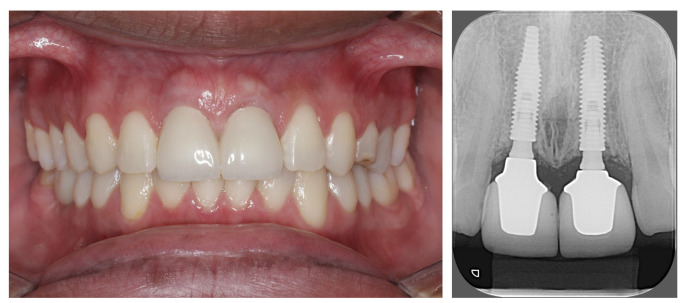
Front view and periapical X-ray showing the stability of the bone tissue.

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
