# Peer review of "Digital Planning for Immediate Implants in Anterior Esthetic Area: Immediate Result and Follow-Up after 3 Years of Clinical Outcome—Case Report"

_dentistry, 2023, doi:10.3390/dj11010015_

Round 1

Reviewer 1 Report

Case report of considerable interest for the dental sector, requires a major revision before proceeding with the evaluation for publication.

Abstracr described correctly

Few keywords, add specific ones

Introduction, it is missing at the end how the microbiota of the implant patient changes in order to plan a right maintenance plan 10.3390 / app12073250

Materials and methods, I suggest to insert the clinical photos in high resolution, the rest are good and it is well described.

Results, highlight more the result obtained

Discussion, to add as future objectives the professional and home management of perio implant maintenance, through the use of minimally invasive systems and natural substances as already studied by the research group of the prof scribante. 10.3390 / microorganisms10040675

10.3390 / app12062800

conclusions add proactive action

bibliography, add required references.

Author Response

Dear Review

Thank for the excellent review 

I will perform the english revision whit the MDPI team the improve the article as soon as the revisions are finished. 

Reviewer 2 Report

Thank you for submitting your work for evaluation. First, The title is not accurate which makes the readers expect to read a detailed and novel approach in treating maxillary incisors with immediate implants. But, neither the guided surgery, the provisional restoration, impression nor the definitive restorations fabrication were adequately described. Also the case has nothing special about the treatment plan sequence to be distinguished from thousands of case reports of the same clinical situation.

Second, the written manuscript has serious flaws in English as well as the quality of scientific writing (Notes were added in the PDF attached for your reference).

Regards

Author Response

dear reviewer 

thank you for the excellent review 

I will perform the English revision with the mdpi team to improve the article as soon as the revisions are finished.

Round 2

Reviewer 1 Report

the manuscript has been properly revised,

Author Response

Dear reviewer 
Thank you very much for the excellent review 

Reviewer 2 Report

Dear Author

Thank you for the effort. Please find the attached comments. Kindly pay attention to the comments ignored from review round 1.

Regards

Author Response

(The authors gave the same response as above.)

Round 3

Reviewer 2 Report

Dear Authors

The manuscript is significantly improved but english language editing is mandatory.